# The Combination of Conventional QTL Analysis, Bulked-Segregant Analysis, and RNA-Sequencing Provide New Genetic Insights into Maize Mesocotyl Elongation under Multiple Deep-Seeding Environments

**DOI:** 10.3390/ijms23084223

**Published:** 2022-04-11

**Authors:** Xiaoqiang Zhao, Yining Niu

**Affiliations:** State Key Laboratory of Aridland Crop Science, Gansu Agricultural University, Lanzhou 730070, China

**Keywords:** constitutive-QTL, bulked-segregation analysis, RNA-sequencing, maize, mesocotyl length, deep-seeding tolerance

## Abstract

Mesocotyl length (MES) is an important trait that affects the emergence of maize seedlings after deep-seeding and is closely associated with abiotic stress. The elucidation of constitutive-QTLs (cQTLs) and candidate genes for MES and tightly molecular markers are thus of great importance in marker-assisted selection (MAS) breeding. Therefore, the objective of this study was to perform detailed genetic analysis of maize MES across 346 F_2:3_ families, 30/30 extreme bulks of an F_2_ population, and two parents by conventional QTL analysis, bulked-segregation analysis (BSA), and RNA-sequencing when maize was sown at the depths of 3, 15, and 20 cm, respectively. QTL analysis identified four major QTLs in Bin 1.09, Bin 3.04, Bin 4.06–4.07, and Bin 6.01 under two or more environments, which explained 2.89–13.97% of the phenotypic variance within a single environment. BSA results revealed the presence of seven significantly linked SNP/InDel regions on chromosomes 1 and 4, and six SNP/InDel regions and the major QTL of qMES4-1 overlapped and formed a cQTL, cQMES4, within the 160.98–176.22 Mb region. In total, 18,001 differentially expressed genes (DEGs) were identified across two parents by RNA-sequencing, and 24 of these genes were conserved core DEGs. Finally, we validated 15 candidate genes in cQMES4 to involve in cell wall structure, lignin biosyntheis, phytohormones (auxin, abscisic acid, brassinosteroid) signal transduction, circadian clock, and plant organ formation and development. Our findings provide a basis for MAS breeding and enhance our understanding of the deep-seeding tolerance of maize.

## 1. Introduction

Maize (*Zea mays* L.) is one of the most important agro-economical crops that is utilized as food, animal feed, and raw material for bioenergy. Successful and continuous production of maize is critical for food security [1,2,3]. Due to adverse environments, such as drought and low temperature stress during maize seed germination, it is essential to develop various strategies to anticipate detrimental changes. Making more resistant maize varieties to be capable of growing in adverse environments is a key strategy to maintain normal growth and high yields [1,4]. Deep-seeding is another practice to anticipate detrimental changes. Unlike the traditional maize sowing depth at 3–5 cm, deep-seeding (>10 cm) is an ancient agricultural practice that can be used to ensure seeds reach the deeper moisture soil, which achieves better seedling emergence and protects seedlings from damage [2,4,5,6]. Deep-seeding has been applied in lots of areas, such as the southwestern area of the USA, parts of western Mexico, and China. Maize varieties of “P1213733 (Komona)” was sown at 30 cm depth or more [7], and the “40107” maize variety was planted to a sowing depth of 26 cm in arid and semi-arid regions of China [4].

When seeds are sown in deep, the ability of organ elongation plays a critical role in plant emergence. Elongating organs differ across various plants. For example, the elongated coleoptile helps deliver the stem and first internode from the embryo to the soil surface in wheat (*Triticum aestivum* L.) [8,9] and barley (*Hordeum vulgare* L.) [10,11], whereas the elongated mesocotyl in rice (*Oryza sativa* L.) plays an important role in pushing the shoot tip across the soil surface during germination [12,13]. The elongated mesocotyl in maize pushes the coleoptile across the soil surface during the germination period, and its growth of mesocotyl is inhibited by light as soon as the coleoptile sprouts from the soil surface [14,15]. Previous studies demonstrated the strong association between mesocotyl length (MES) and seedling vigor (SV; SV refers to the ability and potential of seeds to develop rapidly and emerge in a uniform manner in adverse environmental stimulations); the longer mesocotyl seedlings are better in terms of survival and lead to higher production [16,17,18].

The genetic mechanisms of maize MES are complex and are influenced by genetic backgrounds, population types, population sizes, and environmental factors, including soil texture, temperature, light penetration, and sowing depth [5,6,17]. Thus far, only a limited number of quantitative trait loci (QTLs)/genes have been identified for deep-seeding tolerant traits, especially MES in maize. For example, Han et al. [17] and Liu et al. [5] identified 23 QTLs for MES based on B73 × Mo17 IBM Syn4 recombinant inbred line (RILs; 243 lines)/IBM Syn10 doubled haploid (DH; 280 lines) populations at a sowing depth of 12.5 cm, and six candidate genes controlling the deep-seeding tolerance of maize were identified: *GRMZM2G059167* (*MYB38*), *GRMZM2G133836* (*GA insensitive dwarf 1*, *GID1*), *GRMZM2G139680* (*peroxiredoxin family*), *GRMZM2G140633* (*cell cycle regulator*, *cyclin delta-2*), *GRMZM2G151230* (*cell number regulator 2*, *CNR2*), and *GRMZM2G098460* (*drought-induced protein 1*), in multiple QTLs regions [5]. In another study, Chen et al. [19] identified three linked single nucleotide polymorphisms (SNPs) that were associated with MES in 165 inbred maize lines via genome-wide association studies (GWAS), and three candidate genes: *GRMZM2G149580* (*acyl carrier protein*, *ACP4*), *GRMZM2G037378* (*plus-3 domian-containing protein*), and *GRMZM2G078906* (*FAD binding domain-containing protein*) were further validated in these SNPs regions. In addition, Zhang et al. [6] reported 12 QTLs responsible for MES based on a 3681-4 × X178 F_2:3_ (221 families) population at a sowing depth of 10 and 20 cm, and the major QTL of qMES20-10 was further analyzed using a 170 BC_3_F_3_ population. Our study identified multiple genes involved in cell wall components, programmed cell death, lignin biosynthesis, and a range of phytohormones, including auxin (AUX), ethylene (ET), brassinosteroid (BR), zeatin (ZT), abscisic acid (ABA), gibberellin (GA), jasmonic acid (JA), and salicylic acid (SA), signaling transduction to regulate the development of maize mesocotyls at sowing depths of 3 and 20 cm by RNA-sequencing (RNA-Seq) [2]. However, there has been virtually no progress in the isolation of major QTLs by QTL fine mapping or map-based cloning for marker-assisted selection (MAS) breeding, partly due to the poor resolution of the genetic maps used for research performed prior to the availability of modern marker technology platforms. Therefore, it has been challenging to select maize mesocotyls with higher elongation ability from deep-seeding environments.

Due to recent progress in high-throughput sequencing technology, the bulked-segregation analysis (BSA) method has been widely used to locate genes related to important traits in maize [20], rice [21], and wheat [22], and based on DNA-Seq or RNA-Seq, currently the fastest method for genetic mapping. In this regard, the objectives of the present study were: (i) to identify QTLs controlling MES based on conventional QTL analysis across a 346 F_2:3_ population under sowing environments of 3, 15, and 20 cm depths, (ii) to analyze linked BSA regions responsible for MES via BSA using genome re-sequencing across “30 mixed F_2_ plants with the largest MES”-bulked pool (LM pool), “30 mixed F_2_ plants with the shortest MES”-bulked pool (SM pool), and their two parental inbred lines; and (iii) to explore constitutive-QTLs (cQTLs; i.e., overlapping QTLs regions associated with multiple deep-seeding tolerant traits from both the present research and previous studies), identify and mine the underlying candidate genes in corresponding cQTLs by combining RNA-Seq and quantitative real-time PCR (qRT-PCR) analyses across the two parent mesocotyls at sowing depths of 3 and 20 cm. Our findings provide a basis for further QTL fine mapping, MAS breeding, and functional studies of deep-seeding tolerance in maize.

## 2. Results

### 2.1. MES Variation and Genetic Analysis

Under the same sowing conditions, the MES of the deep-seeding tolerant W64A was significantly larger than the intolerant K12 (*p* < 0.05): 3.82 cm vs. 2.32 cm, 10.28 cm vs. 3.28 cm, and 12.70 cm vs. 5.20 cm, at sowing depths of 3, 15, and 20 cm, respectively (Figure 1C). This data indicated that deep-seeding tolerant maize exhibited longer mesocotyls under the control of its own genetic factors. Moreover, compared to the normal sowing depth of 3 cm, the MES was increased by 169.11, 41.38, 233.77, and 192.78% in W64A, K12, F_1_ hybrid, and F_2:3_ families under 15 cm deep-seeding condition. Meanwhile, it was increased by 232.46, 124.14, 293.77, and 250.00% in these materials under 20 cm deep-seeding condition, respectively (Figure 1E). This data indicated that the significant elongation of mesocotyls determined the tolerance of maize to deep-seeding; thus, mesocotyls play a key role during the emergence of maize seeds in deep layers of soil. Moreover, the mesocotyls of F_1_ hybrid were longer than both parents suggested that it might be when controlled by a limited number of dominant genes and with hybrid vigor involved. Furthermore, the MES of 346 F_2:3_ families under three sowing depth conditions followed a normal distribution, as the absolute value of skewness and kurtosis of the MES was <1.0 (Appendix A). This finding showed that MES was controlled by several QTLs. The estimated broad-sense heritability (*H_B_*^2^) and genotype × environment interaction heritability (*H_GE_*^2^) values of MES in a 346 F_2:3_ population across all environments were 90.96 and 8.98%, respectively (Appendix A). This data demonstrated that MES was less affected by the environment when compared to other deep-seeding tolerant traits, especially emergence rate (RAT) [6]. In addition, 30/30 extreme phenotype families (approximately 8.46% each) from each tail of the MES (mean: 16.21/5.47 cm) were distributed in 346 F_2:3_ families, at a soil depth of 20 cm. We selected their corresponding 30/30 mixed F_2_ plants to form to bulked groups (LM pool and SM pool), which were used for subsequent BSA (Figure 1A,B,D).

### 2.2. Framework of Phenotype and Physiological Metabolisms of MES

We then selected two parents, an F_1_ hybrid and two bulked pools under 3 and 20 cm conditions, to measure the level of lignin and eight phytohormones (Figure 2A). Unlike the normal sowing depth of 3 cm, their average content of lignin and gibberellic acid 3 (GA_3_) decreased by 38.07 and 71.77% under 20 cm deep-seeding condition, respectively; however, their average content indole-3-acetic acid (IAA), cis-zeatin (Cis-ZT), trans-zeatin (Trans-ZT), ABA, 24-epibrassinolide (EBR), JA, and SA increased by 82.78, 156.40, 112.73, 107.55, 170.71, 205.35, and 72.49%, respectively (Figure 2A). In addition, correlation analysis showed that MES had significantly positive correlations in terms of IAA, ABA, EBR, Cis-ZT, Trans-ZT, and SA content, and displayed significantly negative correlations with lignin and GA_3_ content (Figure 2B,C). At the same time, a total of 28 significantly positive/negative correlations were identified among other nine physiological metabolisms (Figure 2B,C). These data indicated that there was close correlation among MES, phytohormones level, and lignin content. Therefore, maize mesocotyl elongation is largely due to changes in these physiological metabolisms at 20 cm sowing depth.

### 2.3. QTLs Analysis of MES across Three Sowing Conditions

Single-environment mapping with composite interval mapping (CIM) was used to investigate the genetic control underlying MES improvement. Seven significant QTLs (*p* < 0.05) for MES in the 346 F_2:3_ population were identified under 3, 15, and 20 cm sowing conditions, and distributed on chromosomes (Chr.) 1, 3, 4, 6, and 7, respectively (Table 1; Figure 3A,B). For these identified QTLs, approximately 68.75, 6.25, 6.25, and 18.75% showed additive (A), partial-dominance (PD), dominance (D), and over-dominance (OD) effects, respectively (Figure 3C). These data showed that MES was regulated by both additive and non-additive effects under different sowing depths. In addition, four major QTLs (phenotypic variance explained [PVE] > 10%) were detected. qMES1-1 (umc2047-bnlg1597 interval) explained 2.89, 8.34, and 10.01% of PVE in the 3, 15, and 20 cm soil conditions, respectively; qMES3-1 (umc1527-umc2261 interval) explained 11.60 and 4.94% of PVE under 15 and 20 cm conditions, respectively; qMES4-1 (umc1869-umc1775 interval) explained 3.81, 9.46, and 13.97% of PVE under the three depth conditions, respectively, and qMES6-1 (umc2311-umc2196 interval) explained 8.11, 5.06, and 10.03% of PVE at the three depth conditions, respectively. These four major QTLs from a deep-seeding tolerant W64A background had positive effects on MES at 15 and 20 deep-seeding conditions (Table 1). These data indicated that these major QTLs increased the MES under deep-seeding stress via the W64A allele.

### 2.4. BSA of MES under 20 cm Sowing Conditions

BSA was used to re-sequence the LM pool, SM pool, W64A, and K12 under 20 cm sowing condition. After screening to remove low-quality reads, we obtained reads of 217,228,830 and 198,744,069, and 65.17 and 59.62 Gb from W64A and K12, respectively (Appendix A). Furthermore, we obtained a total of 187.61 Gb clean data from the two bulked pools (84.71 Gb for the LM pool and 102.90 Gb for the SM pool). After mapping these reads to the *Zea_mays* B73_V4 reference genome, the coverage of the LM pool and SM pool genomes was 41.10× and 50.10×, respectively (Appendix A).

In total, 10,910,818 and 11,030,104 SNPs were detected from W64A and K12, respectively, of which 119,191 and 118,942 SNPs were non-synonymous (Appendix A). At the same time, 15,382,895 and 15,604,761 SNPs were obtained from the LM pool and SM pool, respectively, of which 174,520 and 176,522 SNPs were non-synonymous (Appendix A). In addition, 2,454,187, 2,473,420, 2,664,711 and 2,663,176 small insertion-deletions (InDels) were identified from W64A, K12, LM pool, and SM pool (Appendix A). To identify the linked BSA regions associated with MES, the SNP/InDel index was used to measure the allele segregation of SNPs between the two bulked pools, and calculated using the method described by Takagi et al. [23]. The results showed that three significantly linked SNP regions and four significantly linked InDel regions were responsible for MES (Table 2; Figure 4A,B). For these linked BSA regions, BSA-SNP1-1 was located on Chr. 1 from 243,149,664 to 251,888,006 bp and contained 169 genes; a further six linked BSA regions were relatively close to each other on Chr. 4 from 160,984,132 to 176,222,964 bp and contained 329 genes (Table 2).

### 2.5. Transcriptome Analysis of Two Parent Mesocotyls

Nine RNA-Seq libraries were constructed from the mesocotyls of W64A and K12 under 3 and 20 cm sowing conditions. An average of 7.49 G clean reads with a Q30 value of 94.38% were obtained from each sample (Appendix A) of which there were 50.76 billion high-quality (HQ) clean reads. Then, approximately 88.86% of the HQ clean reads were aligned against the *Zea_mays*. B73_V4 reference genome. Finally, 18,001 DEGs were identified across all comparisons [VS1: W64A (20 cm)_VS_K12 (20 cm); VS2: W64A (3 cm)_VS_K12 (3 cm); VS3: W64A (20 cm)_VS_W64A (3 cm); and VS4: K12 (20 cm)_VS_K12 (3 cm)] (Figure 5A–D). It was interesting to note that 24 core conserved DEGs were further detected during our comparative analyses (Figure 5D,E; Appendix A). It is possible that these core conserved DEGs may be associated with important traits such as mesocotyl elongation that fundamentally differs between the two parents. In addition, the DEGs in each comparison were subjected to KEGG pathway enrichment analysis; the top-20 pathways were “plant hormone signal transduction (map04075)”, “phenylpropanoid biosynthesis (map00940)”, “circadian rhythm-plant (map04712)”, “zeatin biosynthesis (map00908)”, and “porphyrin and chlorophyll metabolism (map00860)” (Figure 5F–I). These data suggested that the differences of MES between W64A and K12 under different sowing environments may be influenced by multiple DEGs in specific KEGG pathways.

### 2.6. cQMES4 Identification and Candidate Genes Analysis in cQMES4

Next, we attempted to obtain reliable cQTLs for deep-seeding tolerant traits to lay a foundation for fine mapping and candidate gene prediction, cloning, and verification. Further analysis showed that in Bin 4.06–4.07, the major QTL of qMES4-1 and six linked BSA regions were highly overlapped and formed a cQTL, i.e., cQMES4 (Figure 6A–C). Furthermore, this interval also contained six other QTLs responsible for MES, coleoptile length (COL), seedling length (SDL), vigor index (VI), and mean germination time (MGT) under 2 and 12.5 cm sowing environments, and explained 54.58% (range 2.97–13.97%) of PVE (Figure 6E). In addition, we used the common genes between RNA-Seq and cQMES4 region to identify 15 candidate genes involved in the deep-seeding tolerance of maize, especially with regards to regulating the elongation of mesocotyls (Figure 6C,D and Appendix A).

### 2.7. qRT-PCR Analysis of Six Candidate Genes in cQMES4

Six candidate genes were selected for expression analysis between W64A and K12 under 3 and 20 cm sowing environments (Figure 6F). Our results showed that all candidate genes showed significant variation in expression (*p* < 0.05) in the W64A/K12 mesocotyls under different sowing environments.

## 3. Discussion and Conclusions

### 3.1. The Relationship of Mesocotyls, Abiotic Stress, and Agronomic Traits

Maize is susceptible to soil water deficit and its yield decreases following cold damage at germination and the early seedling stage, as is typical in agriculture practice. Fortunately, there are abundant genetic variations of mesocotyls among different maize genotypes. Selecting and cultivating maize varieties with longer mesocotyls and stronger deep-seeding tolerance is an important aspect of improving emergence and grain yield in moisture or effective accumulated lower temperature regions [24]. An increasing body of evidence has revealed that the elongation of maize mesocotyls is significantly affected by sowing depths [6], moisture [12], light [24,25,26], and temperature [27]. Under various sowing depths, the phenotyping of MES in 221 maize F_2:3_ families ranged from 10.38–16.89 cm to 10.39–21.50 cm under sowing depths of 10 and 20 cm, respectively, and showed a larger variation at a depth of 20 cm (coefficient of variation, CV: 14.35%) than at a depth of 10 cm (CV: 8.40%) [6]. These findings were consistent with our present results, in that the MES of 346 F_2:3_ family lines were promoted by sowing depth in vermiculite and varied from 0.72 to 5.51 (mean: 3.74 cm; CV: 16.31%), from 3.30 to 15.75 (mean: 10.95 cm; CV 23.47%), and from 4.00 to 19.34 cm (mean 13.09 cm; CV: 44.92%) at depths of 3, 15, and 20 cm in vermiculite, respectively (Appendix A). Then, we measured the contents of lignin and phytohormones across two parents, an F_1_ hybrid, and two bulked pools under 3 and 20 cm conditions. Their correlation also revealed that lignin biosynthesis, as well as phytohormone production and signaling transduction, may play a critical role in the germination of maize seed and elongation of the mesocotyls under deep-seeding stress (Figure 2A–C). The MES also increased with increasing soil water content within a certain soil depth range, but then decreased beyond this range [28]. Therefore, long mesocotyls are important for maize germination, the emergence of seedlings, and the establishment of seedlings under deep-seeding conditions. However, the growth of maize seedlings was adversely affected when seeds were sown deeper than the optimum sowing depth of 3–5 cm [5]. It was interesting to note that previous researchers had initially proved that MES was not correlated with plant height at maturity [12,29]; however, the elongation of mesocotyls in rice could be stably inherited from generation to generation in semi-dwarf varieties [30]. This data implied that it is feasible to select both for MES and ideal plant height and we can aggregate multiple alleles that control for both long mesocotyls and dwarf varieties to form an elite form of maize in the future. Generally, the elongation of mesocotyls in maize is immediately inhibited by light exposure and significantly promoted in darkness. As with rice [12], the influence of different wavelengths on the elongation of mesocotyls in maize followed a specific sequence: darkness > red light > far-red light > blue light [25,26]. Blue light induced the inhibition of mesocotyl growth along with an increase in cell wall stiffness and an increase in extractable lignin in the outer mesocotyl elongation zone in maize [26]. The red-absorbing pigment in maize interacts with phytochrome (PHY) and plays a role in the inhibition of mesocotyl growth. Research has also shown that PHY A can facilitate the inhibition of mesocotyl elongation, and the accumulation of anthocyanin by continuous red and far-red light [25]. Red light has also been shown to inhibit the elongation of the etiolated maize mesocotyls by reducing the supply of AUX from the coleoptile unit [24]. In addition, maize mesocotyls are more easily damaged during chilling stress than other tissues, including the roots, leaves, and coleoptiles [27]. Consequently, mesocotyl is the most sensitive tissue to low-temperature injury during seed germination in maize. In the context of these facts, longer mesocotyl can as a desirable trait in maize, which is an asset for seedling emergence and establishment in adverse environments of top layer of the soil, particularly drought prone regions with less moisture. Furthermore, the genetic insights into MES will provide a foundation for the genetic improvement and optimization of maize genotypes across different environments.

### 3.2. Genetic Dissection of MES in Maize and the Identification of cQTLs via Different Methods

To further determine the genetic basis of deep-seeding tolerance in maize, especially MES, we validated seven QTLs (four, seven, and five QTLs at 3, 15, and 20 cm depths, respectively) responsible for MES cross a W64A × K12 F_2:3_ (346 families) population under three sowing environments by conventional QTL analysis with CIM. Analysis showed that 68.75% and 31.25% of QTLs exhibited both additive and non-additive effects, respectively. These results are consistent with previous findings [31]. It is clear that additive and non-additive effects, as well as the general and specific ability for combination, should be applied to improve the MES of maize under normal and deep-seeding environments. Furthermore, MES can be modified in early generations by simple backcrossing or single cross recombinations. By considering the F_1_ heterosis index (HI, 161.44%), relative heterosis (RH, 36.15%), mid-parent heterosis (MH, 61.44%), over-parent heterosis (OH, 15.05%), and F_2:3_ advantage reduction rate (ARR, −10.43%) of MES, we were able to accurately select the basic background of maize with long mesocotyls and tolerance to deep-seeding environments for our study (Figure 1F). For these identified QTLs, we further identified four major QTLs for MES (Table 1). Two of these imparted pleiotropic effects on multiple deep-seeding tolerance traits. For example, qMES1-1 was mapped in Bin 1.09 with umc2047-bnlg1597 interval (263.01–274.71 Mb) and explained 2.89–10.01% of the PVE under three depth environments. Furthermore, multiple QTLs for RAT (umc1085-umc1969-phi055 interval [6]; near chr01.2532.5 SNP [5]), SDL (umc1085-umc1969 interval [6]), germination index (GI) (near PZE-101221874 SNP [32]), and VI (PZE-101226516-PZE-101229026 interval [32]) also overlapped qMES1-1 under different environments. We also detected qMES4-1 in Bin 4.06–4.07 with umc1869-umc1775 interval; this explained 3.81–13.97% of the PVE under all environments. These data agreed with the QTL results for MES (near chr04.1595.5 [5]; close to umc1775 [17]), COL (near chr04.1699 [5]), SDL (close to chr04.1623.5 [5]), VI (near PZE-104087575 SNP [32]), and MGT (near PZE-104087575 [32]) in multiple environments. These findings revealed that the two major QTLs in Bin 1.09 and Bin 4.06–4.07 control traits that are closely linked and related to deep-seeding tolerance. This also explains why the loci of these pleiotropic major QTLs are associated with a larger PVE. In addition, we have to point out that MES phenotyping of the F_2:3_ families may not provide accurate information on the MES phenotype of the corresponding F_2_ population, thus the QTLs analysis in this study may not be as accurate. To more objectively evaluate the MES phenotype and map reliable QTLs, it is necessary to construct RIL or DH population for further study in the future.

It is well known that BSA is an efficient method for reducing genome complexity and is also a rapid, cheap, and highly suitable method for mapping different populations. Therefore, we also combined BSA with re-sequencing and used this to rapidly identify specific genomic regions for MES in maize. Seven linked SNP/InDel regions were associated with MES and mapped to Chr. 1 (243.15~251.89 Mb) and 4 (160.98~176.22 Mb); these featured 169 and 329 genes, respectively (Table 2). In addition, we noted that six BSA regions overlapped with the major QTL, i.e., qMES4-1 in Bin 4.06–4.07, which formed cQMES4 (Figure 6B,C); previous research identified six other QTLs for multiple deep-seeding traits in the cQMES4 region [5,17,32] (Figure 6E). Consequently, the presence of cQMES4 in broad genetic backgrounds under adverse environments will provide further guidance for fine mapping and MAS in the future.

### 3.3. Validation of Candidate Genes in the cQMES4 Region and Their Functions Analysis

Next, we considered the DEGs identified by four comparisons with RNA-Seq analysis and annotated these DEGs in terms of the cQMES4 region. We identified 15 candidate genes in the cQMES4 region that are associated with the elongation of mesocotyls in maize. This information was used for the map-based cloning of MES in relation to genetic activity in multiple environments (Figure 6C,D; Appendix A).

GO annotation showed that *Zm00001d051554* (cytochrome P450, 2.49-fold down-regulated in VS2) was involved in various biological functions including multicellular organism development (GO:0007275), ABA metabolic process (GO:0009687), BR homeostasis (GO:0010268), BR biosynthetic process (GO:0016132), and ABA catabolic process (GO:0046345). Similarly, *Zm00001d020418*, a gene encoding a putative cytochrome P450, was shown to be associated with lignin biosynthesis and expressed at significantly higher levels during the elongation of mesocotyls in maize under deep-seeding conditions [2]. *Zm00001d051577* (Dof zinc finger protein DOF2.1, 10.98-/10.59-fold up-/down-regulated in VS2/VS3) was shown to be involved in procambium histogenesis (GO:0010067) and phloem or xylem histogenesis (GO:0010087). *Zm00001d051569* (2.53-/4.11-fold up-/down-regulated inVS2/VS3) was shown to regulate response to blue light (GO:0009637), flower development (GO:0009908), and the positive regulation of flower development (GO:0009911). *Zm00001d051687* (rho GDP-dissociation inhibitor 1, 2.93-fold up-regulated in VS3) was found to be responsible for cell tip growth (GO:0009932) and the differentiation of epidermal cells in the roots (GO:0010053). *Zm00001d051792* (8.10-/6.03-fold down-regulated in VS1/VS2) was associated with entrainment of the circadian clock (GO:0009649) and response to UV-B (GO:0010224). Previous studies showed that PHY A mediates a number of responses, including the inhibition of hypocotyls in *Arabidopsis* under varying light quantities [33]. In a previous study, Dubois et al. [15] used QTL mapping to demonstrate that *PHY B2* was located in Bin 9.03 and controlled both MES and ABA accumulation in maize. *Zm00001d051787* (3-ketoacyl-CoA synthase 5, 2.76-fold up-regulated in VS2) was found to play a role in the response to cold (GO:0009409) and the response to light stimulus (GO:0009416). The function of *Zm00001d051787* was similar to that of *Zm00001d032948*, a stress-induced gene that is expressed in the seeds, roots, and leaves of maize [34]. *Zm00001d051800* (caleosin-related protein, 1.51-fold up-regulated in VS2) was shown to possess plant seed peroxidase (POD) activity (GO:1990137) and oxidoreductase activity (GO:0016491). POD, as a lignin synthetase, plays an important role in dehydrogenation and participates in the polymerization of monolignols to form the cell wall [2]. POD activity is known to increase notably during the accumulation of lignin in the roots of rice; furthermore, root lignification has been shown to reduce growth levels under copper stress [35]. It is possible that *Zm00001d051802* (auxin-responsive protein SAUR50, 6.40-fold up-regulated in VS2) was found to perform the same function as *AtSAUR24* in *Arabidopsis* in regulating the development of mesocotyls [36]. *Zm00001d051829* (U-box domain-containing protein 44, 1.75-fold down-regulated in VS2) was shown to play a role in the regulation of ABA biosynthesis (GO:0010115), leaf senescence (GO:0010150), the regulation of catabolism in chlorophyll (GO:0010271), the regulation of the biosynthesis of chlorophyll (GO:0010380), and the negative regulation of ABA biosynthesis (GO:0090359). *Zm00001d051837* (sucrose synthase 6, 1.30-fold down-regulated in VS2) was found to participate in sucrose metabolism (GO:0005985) and the synthesis of cellulose [37]. A previous study showed that overexpression of *Populus* L. sucrose synthase in tobacco resulted in a thickened cell wall and increased height [37]. *Zm00001d051847*, i.e., *Nana2-like 1* (1.16-fold up-regulated in VS3) contains the FAD-binding domain and plays a role in the lignin metabolic process (GO:0009808), unidimensional cell growth (GO:0009826), plant-type secondary cell wall biogenesis (GO:0009834), and BR biosynthetic process (GO:0016132). *GRMZM2G078906* (*FAD binding domain-containing protein*), and was also identified as close to the PZE-110040848 marker responsible for the MES and seed germination in maize [19]. Du et al. [38] confirmed that *ZmMYB59* negatively regulated the germination ability of maize seed. In this study, we identified a homologous *Zm00001d051520* (-2.35-fold down-regulated in VS2) in the *MYB* family. *Zm00001d051529* (2.92-fold up-regulated in VS2) was found to possess 4-coumarate-CoA (4CL) ligase activity (GO:0016207) and therefore plays a role in the phenylpropanoid metabolic process (GO:0009698). Zhao et al. [2] identified five DEGs for 4CL in different cultures of maize mesocotyls. *Zm00001d051925* (4.54-fold up-regulated in VS2) played a role in cell surface receptor signaling pathways (GO:0007166), protoderm histogenesis (GO:0010068), cell differentiation (GO:0030154), cuticle development (GO:0042335), and plant organ formation (GO:1905393), whereas *Zm00001d051928* (glutathione S-transferase family protein, 2.26-fold down-regulated in VS2) had effect the negative regulation of auxin-mediated signaling pathways (GO:0010930).

In conclusion, these 15 candidate genes in cQMES4 region can be used as an excellent source of candidate genes for investigating the genetic control of maize seed germination and seedling growth under deep-seeding condition. Little is known about the specific roles and functions of these candidate genes in maize, and the *Arabidopsis* model can be used as a highly suitable model for investigating MES and mesocotyl elongation. In future study, our research will focus on the functional verification and signal transduction pathways of these candidate genes.

## 4. Materials and Methods

### 4.1. Development of Maize Populations

An F_2_ population consisting of 346 plants derived from an F_1_ hybrid (W64A × K12) plant was used to construct a genetic linkage map. One of the parent plants (W64A, ♀) had deep-seeding tolerance whereas the other parent plant (K12, ♂) was intolerant [2,14]. We obtained 346 F_2:3_ families by self-crossing all F_2_ plants and then evaluated MES across all F_2:3_ families after 10 days of culture with vermiculite at sowing depths of 3, 15, and 20 cm in a greenhouse, respectively; we did this so that we could identify specific QTLs for MES. According to the MES phenotypes of all F_2:3_ families under the 20 cm condition, we selected 30, and 30 families with the largest and shortest MES, respectively. Next, we found their corresponding F_2_ plants to establish two groups of plants: a “30 mixture F_2_ plants with the largest MES”-bulked pool (LM pool) and a “30 mixture F_2_ plants with the shortest MES”-bulked pool (SM pool) (Figure 1A). We also used individual W64A and K12 mesocotyls to establish parental pools for BSA-sequencing analysis.

### 4.2. Phenotypic and Physiological Evaluation of Mesocotyls

The mesocotyls of W64A, K12, F_1_ hybrid, and 346 F_2:3_ families were examined in a greenhouse in PVC tubes under 3, 15, and 20 cm sowing conditions. First, the seeds of each F_2:3_ family, F_1_ hybrid, and two parental lines were soaked at room temperature (2 °C) for 24 h in 100 mL of distilled water. Then, PVC tubes (height: 50 cm; diameter: 17 cm) were loaded with the evenly-mixed vermiculite to a specified depth and 30 soaked seeds were sown evenly onto the surface. Next, a depth of 3-, 15-, and 20 cm- mixed vermiculite was used to cover the seeds, thus resulting in a total depth of 50 cm. Finally, the PVC tubes were cultured in a greenhouse (22 ± 1.0 °C with a 12/12 h light/dark cycle at 65% relative humidity) for 10 days. A volume of 20 mL of distilled water was poured onto the vermiculite at 2-day intervals. Each sowing condition had three replicates. Subsequently, we took the seedlings out to measure the MES of ten seedlings. The MES of the F_2:3_ families, F_1_ hybrid, and both parental lines under each condition were compared statistically using IBM-SPSS Statistics version 19.0 (SPSS Inc., Chicago, IL, USA) (http://www.ibm.com/products/spss-statistics (accessed on 6 April 2022)). The significance of the total and residual variances of MES in the F_2:3_ families under all conditions were estimated by a general linear model for univariate data and by one-way analysis of variance (ANOVA). Values of *H_B_*^2^ and *H_GE_*^2^ for MES among all conditions were estimated by Equations (1) and (2), as follows [39]:*H_B_*^2^ = σ_g_^2^/(σ_g_^2^ + σ_ge_^2^/n + σ_ε_^2^/nr),(1)
*H_GE_*^2^ = (σ_g_^2^/n)/(σ_g_^2^ + σ_ge_^2^/n + σ_ε_^2^/nr),(2)
where, σ_g_^2^ was the genotypic variance; σ_e_^2^ was the environmental variance; σ_ε_^2^ represented the error variance, σ_ge_^2^ was the variance of genotype × environment interaction; n was the number of sowing depth environments (n = 3); and r was the number of replications (r = 10). The heterosis for MES, including HI, RH, MH, OH, and ARR, were determined by Equations (3)–(7), as follows [40]:HI = F_1_/MP × 100%,(3)
RH = (F_1_ − MP)/F_1_ × 100%,(4)
MH = (F_1_ − MP)/MP × 100%,(5)
OH = (F_1_ − P_H_)/P_H_ × 100%,(6)
ARR = (F_2:3_ − F_1_)/F_1_ × 100%,(7)
where, MP represented the mean MES of both parental lines and P_H_ represented the longest MES of the parental line. Relative to the normal 3 cm condition, the mean rate of change (RC) for MES under the 15 or 20 cm conditions was estimated by Equation (8), as follows [41]:RC = (1 − T_T_/T_N_) × 100%,(8)
where, T_T_ and T_N_ represented the mean value of MES at 3 cm and the 15/20 cm conditions, respectively.

Next, 0.5 g of mesocotyls from the W64A, K12, F_1_ hybrid, LM pool, and SM pool under the 3 and 20 cm conditions were homogenized in 5 mL of ethanol (95%, *v*/*v*) and then centrifuged at 10,000 rpm at 4 °C for 10 min. Then, the sediment was rinsed three times with ethanol-n-hexane solution (1/1, *v*/*v*) and dried. Next, the sediment was dissolved in 0.5 mL of bromide acetyl-glacial acetic acid solution (1/3, *v*/*v*) and then bathed in water for 30 min at 70 °C before being mixed with 0.9 mL of NaOH (2 M), 5 mL of glacial acetic acid, and 0.1 mL of hydroxylamine hydrochloride (7.5 M), to analyze lignin content [4]. Next, 0.5 g of mesocotyls was ground in liquid nitrogen and digested in 5 mL of methanol-formic acid solution (99/1, *v*/*v*) for 12 h at 4 °C and then centrifuged at 12,000 rpm at 4 °C for 20 min; then, the supernatant was collected. The residue was further digested in 5 mL of methanol-formic acid solution (99/1, *v*/*v*) and re-centrifuged, as described above. Then, the supernatants were pooled. Next, we removed pigmentation with a Cleanert ODS C18 solid phase extraction column (Tianjin Aiger Co., Ltd., China), which was then dried by nitrogen flow at 25 °C and then dissolved using 1 mL of methanol. Finally, the solution was filtered with a 0.22 μM membrane filter and 5 μL was injected for analysis. The levels of IAA, ABA, GA_3_, Trans-ZT, Cis-ZT, SA, and JA were analyzed by high performance liquid chromatography (HPLC), and EBR was analyzed by ultra-high performance liquid chromatography-tandem mass spectrometry (UHPLC-MS/MS) [4].

### 4.3. SSRs Analysis, Genetic Linkage Map Contruction, and QTL Analysis

Leaves were collected from each F_2_, W64A, K12, and F_1_ hybrid seedling and stored at −80 °C. High-quality genomic DNA was extracted from each sample using the cetyltrimethylammonium bromide (CTAB) method [41]. SSR analysis was conducted as described by Zhao et al. [41]. In total, 260 polymorphic SSRs were identified between W64A and K12 from the Maize GDB (http://www.maizegdb.org/ (accessed on 22 January 2020)); these were then used to genotype the 346 F_2_ plants. Next, we constructed a 1,410.6 cM genetic linkage map using JoinMap version 4.0 (https://www.kyazma.nl/index.php/JoinMap/ (accessed on 16 June 2021)), with a mean interval of 5.58 cM (Figure 3A).

QTL analysis was performed on MES data in 346 F_2:3_ families under each sowing condition using CIM with Windows QTL Cartographer software version 2.5 (http://statgen.ncsu.edu/qtlcart/WQTLCart.htm (accessed on 20 July 2021)). Model 6 of the Zmapqtl module was used to analyze QTLs by CIM. The window size was 10 cM and cofactors were selected by forward and backward regressions, with in and out thresholds at *p* < 0.05. A genome-wide critical threshold value was estimated for an experimental type I error rate of 0.05 using 1000 random permutations. The genetic action of each QTL was analyzed in accordance with the methods described by Stuber et al. [42] as follows: additive (A; |dominance/additive| = 0.00~0.20), partial-dominance (PD; |dominance/additive| = 0.21~0.80), dominance (D; |dominance/additive| = 0.81~1.20), and over-dominance (OD; |dominance/additive| > 1.20).

### 4.4. BSA-Sequencing and Sequence Alignment

The two DNA F_2_-bulked pools were collected for library construction by mixing equal amounts of DNA from the LM pool with 30 of the largest MES, and from the SM pool with 30 of the shortest MES. Two parental lines, W64A and K12, were also prepared for library construction. Next, the DNA samples from the two F_2_-bulked pools and the two parental lines were prepared according to the standard Illumina protocol, to construct sequencing libraries which were then sequenced on an Illumina NovaSeq 6000 platform (Illumina, San Diego, CA, USA). Illumina Casava 1.8 was used for cleaning and filtering reads [43]. After low-quality and short reads were filtered out, the filtered short reads of each pool were mapped onto the Zea_mays.B73_V4 reference genome sequence (ftp://ftp.ensemblgenomes.org/pub/plants/release-46/fasta/zea_mays/dna/ (accessed on 6 September 2021)) with the Burrows-Wheeler Aligner (BWA) [44]. Sequence Alignment/Map (SAM) tools and the Genome Analysis Toolkit (GATK; https://www.broadinstitute.org/gatk/guide/best-practices?bpm=DNAseq#variant-discovery-ovw (accessed on 10 September 2021)) were used to ensure the accuracy of the SNPs, and potential PCR duplications were removed using the SAM tools command “rmdup” [44]. The Unified Genotyper function in GATK software was used to identify SNPs/InDels variants. Then, the obtained SNPs/small InDels were annotated and predicted using snpEff software (http://pcingola.github.io/SnpEff/ (accessed on 13 September 2021)) [45]; these were then used for BSA analysis.

The depth of the reads for homozygous SNPs/InDels in the offspring pools was then acquired and used to calculate the SNP/InDel index, as described previously by Takagi et al. [23]. The ΔSNP-index was simulated for more than 10,000 replicates for each bulk, based on the population type and size. Then, quantiles (99%) from the simulations were used to estimate confidence intervals. An actual ΔSNP-index averaged over a sliding window that lay outside of the confidence interval indicated a linked region related to MES. Then, we applied Blast software to perform deep annotation in multiple databases: Nr (http://www.ncbi.nlm.nih.gov/pubmed (accessed on 10 November 2021)), GO (http://bioinfo.cau.edu.cn/agriGO/ (accessed on 10 November 2021)), KEGG (http://www.genome.jp/kegg/ (accessed on 10 November 2021)), COG (https://www.ncbi.nlm.nih.gov/COG/ (accessed on 10 November 2021)), and Swiss-Prot (https://web.expasy.org/docs/swiss-prot_guideline.html (accessed on 10 November 2021)), using the coding genes in the candidate interval.

### 4.5. RNA-Seq and DEGs Analysis

Total RNA from W64A and K12 mesocotyls under the 3 and 20 cm sowing conditions (with three biological replicates) was extracted using commercial kits (TRIZOL reagent, Invitrogen, Carlsbad, CA, USA). DNA impurities in the total RNA were digested using DNase. Eukaryotic mRNA was then enriched using magnetic beads with Oligo (dT) and mRNA was fragmented using an interrupting reagent to construct a library using an Agilent 2100 Bioanalyzer (Agilent Technologies, Santa Clara, CA, USA). Then, the resulting ligation products were size-selected by agarose gel electrophoresis, PCR amplified, and sequenced using an Illumina NovaSeq PE150 sequencer to generate 150 bp paired-end reads.

Raw reads were filtered by fastp (v. 0.18.0) to remove adapters and low-quality reads. The cleaned reads were then mapped to the Zea_mays B73_V4 reference genome using Tophat2. The mapped reads of each sample were then assembled by Cufflinks. The expression levels of the merged transcripts were then counted and FPKM values were calculated. DEG analysis was performed by DESeq R software in Bioconductor (http://www.bioconductor.org/ (accessed on 4 January 2021)), and significant DEGs for each comparison were identified using a false discovery rate (FDR) < 0.001 and |log2 fold-change (FC)| > 1.

### 4.6. cQTL Detection and the Dissection of Candidate Genes

We selected corresponding QTL information from our results via QTL and BSA, as well as other original QTLs for deep-seeding tolerant traits under different environments from public databases, including MaizeGDB (http://www.maizegdb.org (accessed on 16 January 2022)), NCBI (http://www.ncbi.nlm.nih.gov (accessed on 16 January 2022)), and CNKI (http://www.cnki.net (accessed on 16 January 2022)). The cQTLs refer to overlapping regions combining multiple QTLs resulting from different sowing depth environments [39]. Then, we combined the DEGs in different comparisons by RNA-Seq and the genes in the cQTLs regions. The common genes between RNA-Seq and cQTLs regions represented candidate genes that play roles in the elongation of mesocotyls. A physical map of cQTLs information and the candidate genes in the cQTLs regions was created by BioMercator v. 4.2 ssoftware (http://www.bioinformatics.org/mqtl/wiki/ (accessed on 26 February 2022)) [46].

### 4.7. qRT-PCR Analysis

We extracted total RNA from the mesocotyls of W64A, K12, F_1_ hybrid, LM pool, and SM pool under both 3 and 20 cm sowing conditions again, which were used for qRT-PCR analysis. The qRT-PCR was performed using a super real premix plus (SYBR Green) (Tiangen, Shanghai, China) in a total volume of 20 μL. The primers used to amplify the six candidate genes as listed in Table 3. The 2^−∆∆Ct^ method was used to determine the relative gene expression levels; these were normalized to the *ZmActin1* gene [2,4,5].

## Figures and Tables

**Figure 1 ijms-23-04223-f001:**
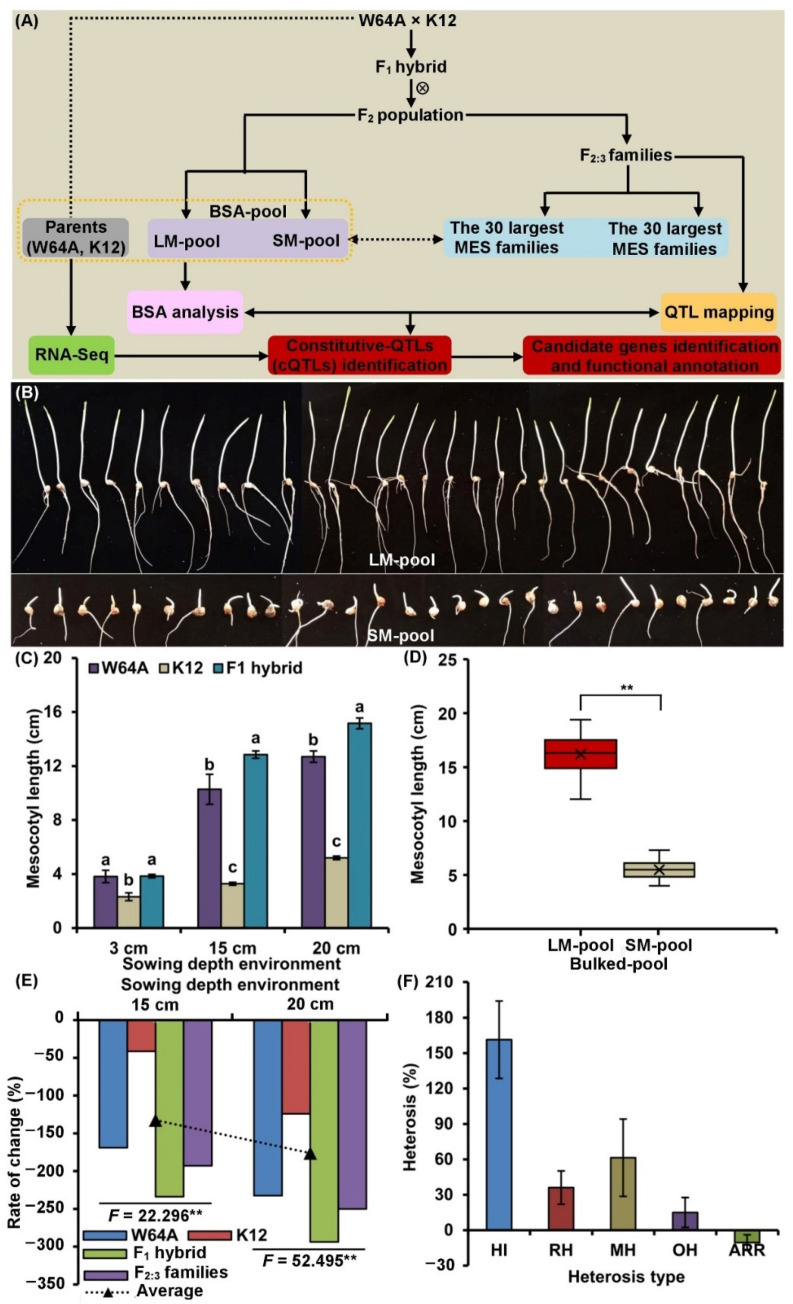
Performance of mesocotyls at three sowing environments. (**A**) Schematic diagram of experimental population construction and analysis. (**B**) Performance of the mesocotyl length (MES) of LM pool (the 30 largest MES-pool) and SM pool (the 30 shortest MES-pool) at a 20 cm sowing environment. (**C**) MES of the parents (W64A and K12) and their F_1_ hybrid under 3, 15, and 20 cm sowing environments, respectively; different lowercase letters with different materials in a single environment indicated a significant difference with *p* < 0.05 (ANOVA). (**D**) Boxplot of MES showing the difference between LM pool and SM pool used for a BSA under 20 cm sowing environment; asterisks indicated significant differences (** *p* < 0.01; ANOVA). (**E**) Compared to 3 cm sowing conditions, the rate of change (RC) of MES in two parents, F_1_ hybrid, and F_2:3_ population under 15 and 20 cm sowing conditions, respectively; asterisks indicated significant differences (** *p* < 0.01; ANOVA). (**F**) Heterosis analysis (HI: F_1_ heterosis index, RH: relative heterosis, MH: mid-parent heterosis, OH: over-parent heterosis, ARR: F_2:3_ advantage reduction rate) of MES.

**Figure 2 ijms-23-04223-f002:**
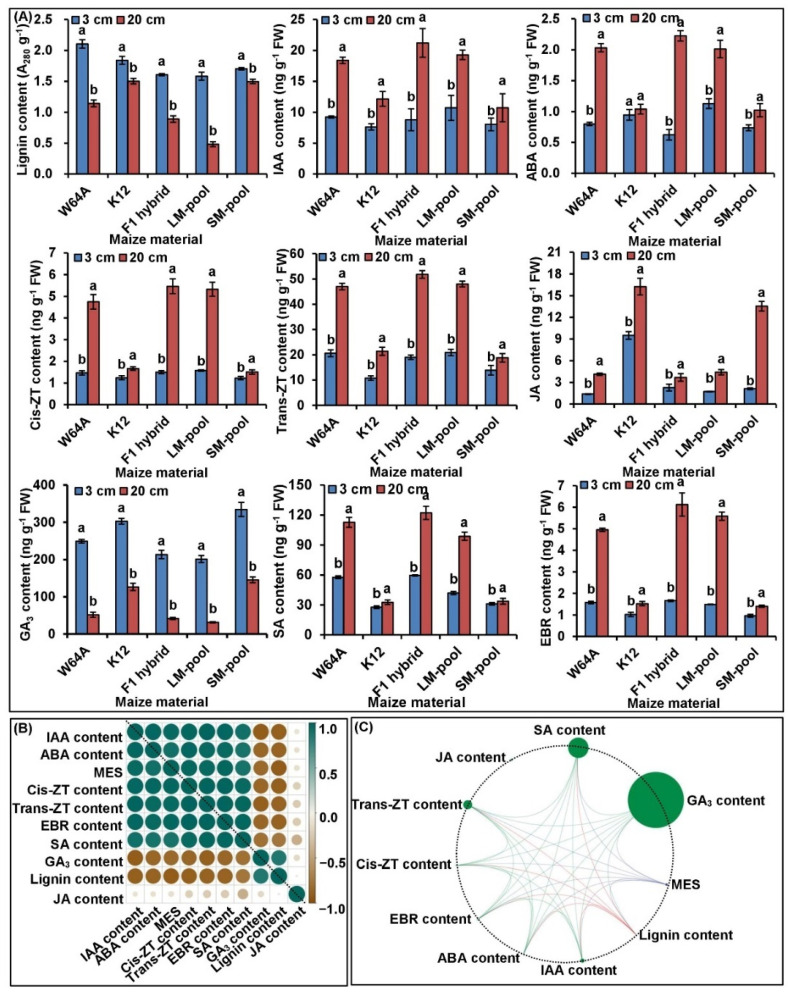
Nine physiological traits assay and the correlation relationship between all physiological traits and mesocotyl length (MES) from W64A and K12. (**A**) Statistics of lignin content, indole-3-acetic acid (IAA) content, abscisic acid (ABA) content, gibberellic acid 3 (GA_3_) content, 24-epibrassinolide (EBR) content, cis-zeatin (Cis-ZT) content, trans-zeatin (Trans-ZT) content, jasmonic acid (JA) content, and salicylic acid (SA) content from mesocotyls of five maize materials under 3 and 20 cm sowing environments; different lowercase letters with a single maize material under different environments indicated a significant difference with *p* < 0.05 (ANOVA). (**B**,**C**) Correlation coefficient diagram and interactive ring correlation diagram (which were prepared using the Genescloud tool; https://www.genescloud.cn (accessed on 6 April 2022)), among these, nine traits of maize materials under different environments).

**Figure 3 ijms-23-04223-f003:**
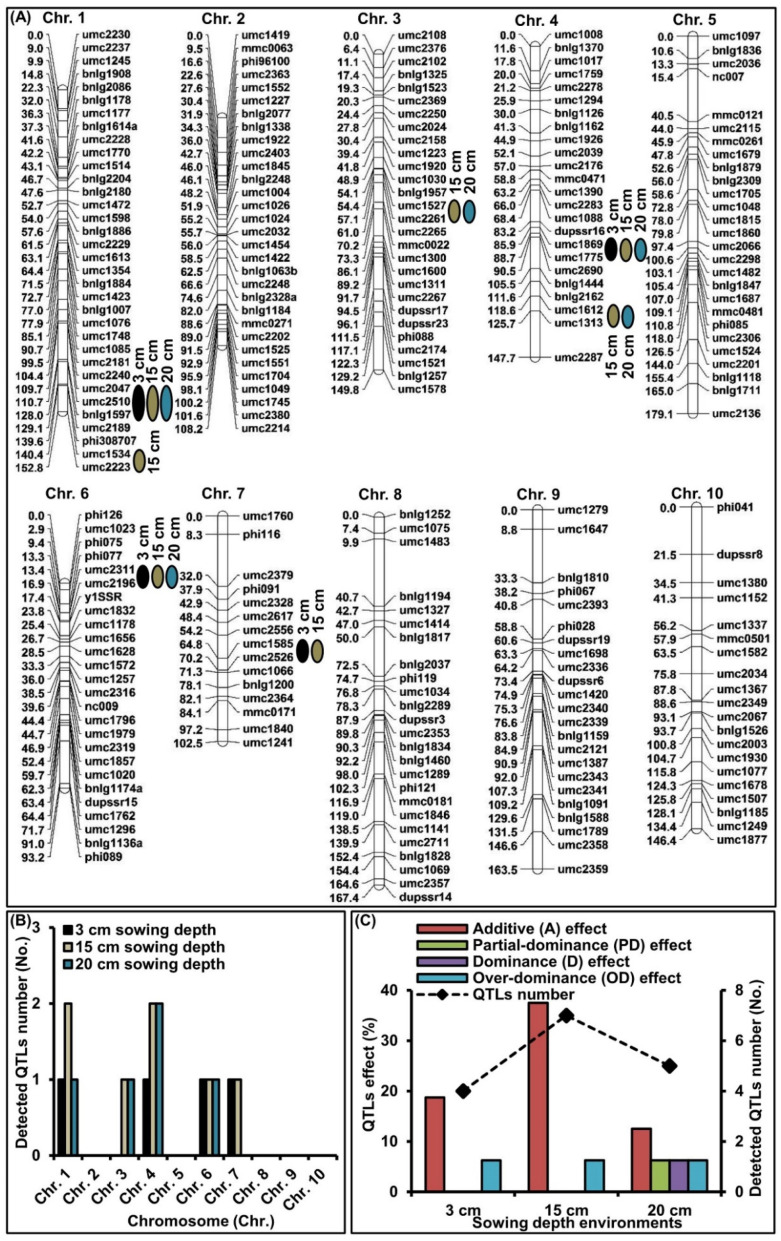
Genetic linkage map based on the F_2_ population and QTLs detected for mesocotyl length (MES) in the F_2:3_ population via single-environment mapping, with composite interval mapping (CIM) under 3 cm, 15 cm, and 20 cm sowing environments, respectively. (**A**) Location of detected QTLs for MES on the genetic map. (**B**) Distribution of detected QTLs on ten chromosomes. (**C**) Gene action analysis of detected QTLs.

**Figure 4 ijms-23-04223-f004:**
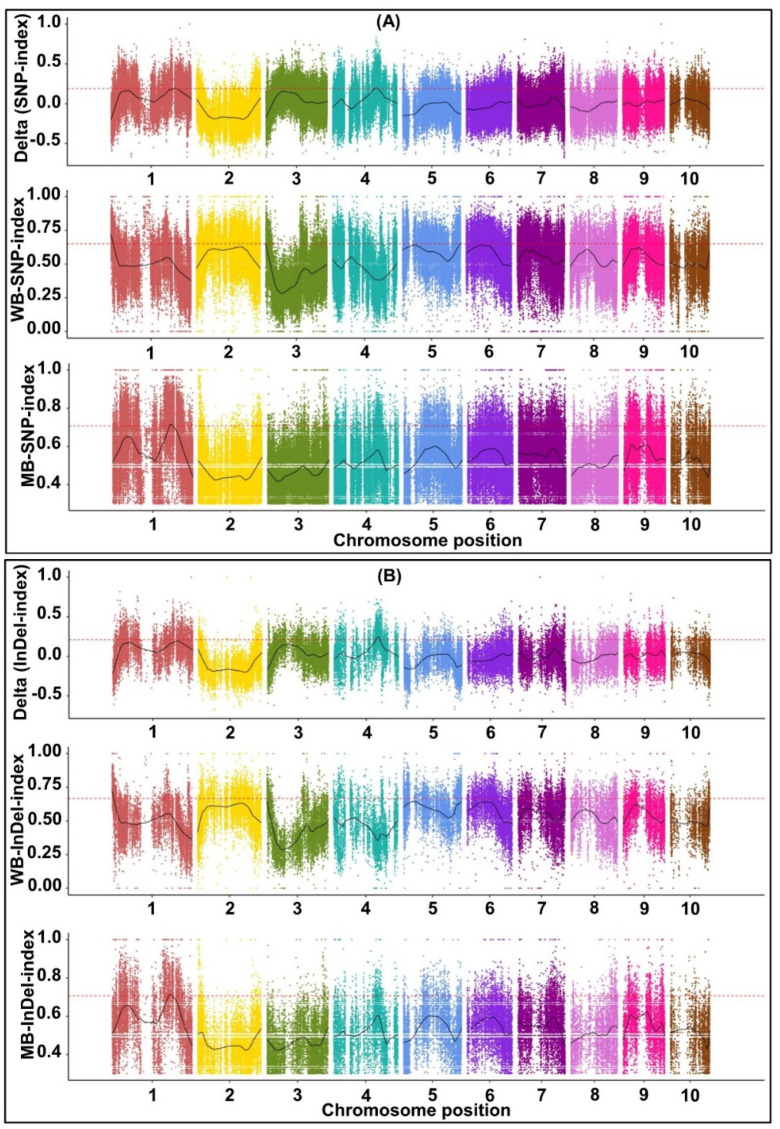
Distribution of SNP index (**A**) and InDel index (**B**) correlation value on chromosomes by BSA analysis of mesocotyl length (MES) under 20 cm sowing environment. The *x*-axis was the chromosomes, the colored dots represented the calculated SNP-index/InDel-index (or △SNP-index/△InDel-index) value, and the black line was the fitted SNP-index/InDel-index (or △SNP-index/△InDel-index) value. The red line represented the threshold line with a confidence level of 0.99.

**Figure 5 ijms-23-04223-f005:**
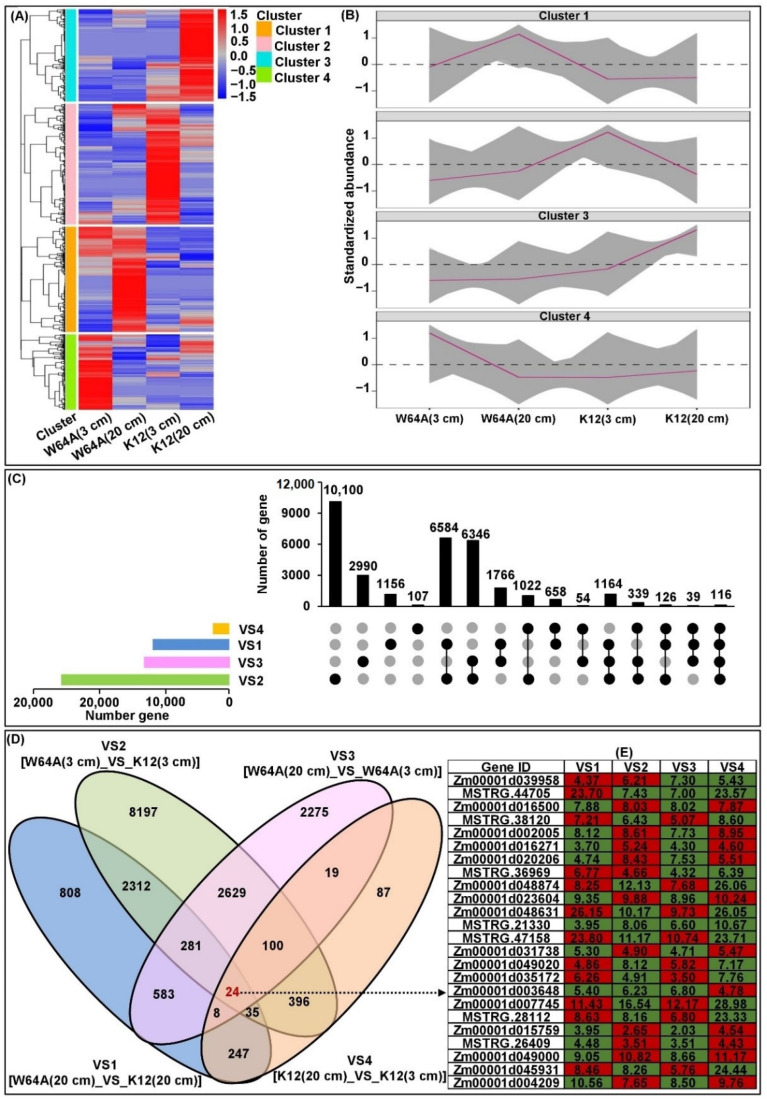
Transcriptional levels of maize mesocotyls at two sowing environments. (**A**) Based on the log2 fold-change value of the comparisons, a hierarchical clustering of genes was performed with W64A and K12 for two conditions to obtain four clusters. (**B**) The representative expression patterns of four clusters were screened. (**C**) All differentially expressed genes (DEGs) were analyzed in VS1 [W64A (20 cm)_VS_K12 (20 cm)], VS2 [W64A (3 cm)_VS_K12 (3 cm)], VS3 [W64A (20 cm)_VS_W64A (3 cm)], and VS4 [K12 (20 cm)_VS_K12 (3 cm)] comparisons; the bar charts indicated DEGs in a single comparison (the *x*-axis represents the number of genes); column charts indicated DEGs in single or multiple comparisons (in the *x*-axis, black dots represented a single comparison; black lines connected by dots represented multiple comparisons; and the *y*-axis represented the number of corresponding genes). (**D**) DEGs identified among four comparisons and 24 core conserved DEGs obtained in all comparisons. (**E**) The expression patterns of 24 core conserved DEGs in four comparisons; the red-box or green-box representing the corresponding DEG was up-regulated or down-regulated, and the value in the box was the log2 fold-change of the corresponding DEG. (**F**–**I**) KEGG pathways enriched in DEGs in VS1, VS2, VS3, and VS4 comparisons, respectively.

**Figure 6 ijms-23-04223-f006:**
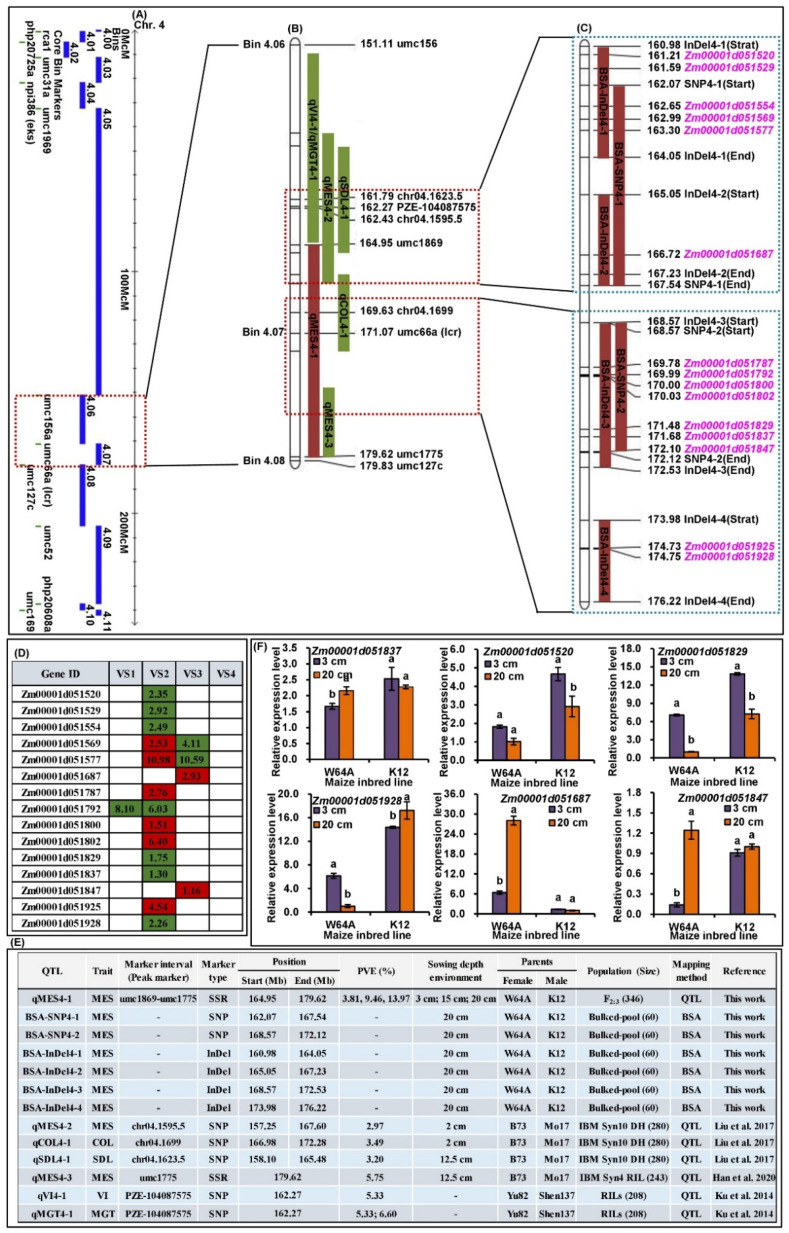
cQMES4 region and corresponding candidate genes information. (**A**) Distribution of different Bins on maize chromosome 4. (**B**) Physical map information of cQMES4 that contained multiple overlapped QTLs for mesocotyl length (MES), coleoptile length (COL), seedling length (SDL), vigor index (VI), and mean germination time (MGT). (**C**) Physical map information of cMES4 that including our six linked BSA regions and 15 identified candidate genes in cQMES4. (**D**) The expression patterns of 15 identified candidate genes in cQMES4 by RNA-sequencing; the red-box or green-box representing the corresponding gene was up-regulated or down-regulated, and the value in the box was the log2 fold-change of the corresponding gene. (**E**) Summary of all QTLs and linked BSA regions information in cQMES4 region from our work and previous studies. (**F**) Relative expression level of six candidate gens in W64A and K12 mesocotyls under 3 and 20 cm sowing environments; different lowercase letters with a single parent under different environment indicated a significant difference with *p* < 0.05 (ANOVA).

**Table 1 ijms-23-04223-t001:** QTLs detected for mesocotyl length (MES) in the F_2:3_ population via single-environment mapping with composite interval mapping (CIM) under 3 cm, 15 cm, and 20 cm sowing environments, respectively.

QTL	Bin	Sowing Depth Environment	QTL Position		LOD	Additive (a)	Dominance (d)	Gene Action		PVE (%)
			cM	Marker Interval				|d/a|	Type	
qMES1-1	1.09	3 cm	110.9	umc2047-bnlg1597	3.10	0.25	0.03	0.12	A	2.89
		15 cm	110.8	umc2047-bnlg1597	8.84	−0.75	−0.11	0.10	A	8.34
		20 cm	110.8	umc2047-bnlg1597	10.43	−0.91	−0.15	0.16	A	10.01
qMES1-2	1.10	15 cm	145.2	umc1534-umc2223	13.96	−1.17	0.10	0.09	A	13.12
qMES3-1	3.04	15 cm	55.3	umc1527-umc2261	7.10	−1.88	−0.21	0.11	A	11.60
		20 cm	56.1	umc1527-umc2261	4.77	−1.03	−1.20	1.17	D	4.94
qMES4-1	4.06–4.07	3 cm	88.1	umc1869-umc1775	4.08	−0.34	−0.02	0.06	A	3.81
		15 cm	88.0	umc1869-umc1775	10.02	−0.83	−0.09	0.11	A	9.46
		20 cm	88.2	umc1869-umc1775	14.84	−1.25	−0.18	0.14	A	13.97
qMES4-2	4.05–4.08	15 cm	119.1	umc1612-umc1313	4.44	−1.10	0.08	0.07	A	6.98
		20 cm	119.4	umc1612-umc1313	4.61	0.17	0.10	0.59	PD	8.42
qMES6-1	6.01	3 cm	14.3	umc2311-umc2196	9.30	−1.01	−1.51	1.50	OD	8.11
		15 cm	14.3	umc2311-umc2196	5.81	−0.61	0.97	1.59	OD	5.06
		20 cm	14.3	umc2311-umc2196	11.45	−1.14	−1.43	1.25	OD	10.03
qMES7-1	7.02	3 cm	67.0	umc1585-umc2526	3.08	−0.80	−0.03	0.04	A	4.93
		15 cm	66.9	umc1585-umc2526	2.29	−0.61	0.08	0.13	A	3.67

MES, mesocotyl length; LOD, logarithm of odds; PVE, phenotypic variance explained by the QTL; Additive (a) effect positive values indicate K12 carries the allele and contributes to an increase in the mesocotyl length, whereas negative values indicate that W64A carries the allele and contributes to an increase in mesocotyl length.

**Table 2 ijms-23-04223-t002:** Mapping candidate SNP-region and InDel-region information based on BSA strategy of mesocotyl length (MES) under 20 cm sowing environment.

Linked SNP-/InDel-Region	Chromosome	Start (bp)	End (bp)	Region Length (Mb)	Genes Number	Unique Genes Number
BSA-SNP1-1	1	243,149,664	251,888,006	8.74	169	169
BSA-SNP4-1	4	162,067,875	167,544,471	5.48	131	329
BSA-SNP4-2	4	168,572,722	172,116,409	3.54	82	
BSA-InDel4-1	4	160,984,132	164,049,553	3.07	64	
BSA-InDel4-2	4	165,049,976	167,225,404	2.18	65	
BSA-InDel4-3	4	168,572,674	172,533,611	3.96	96	
BSA-InDel4-4	4	173,979,835	176,222,964	2.24	74	

**Table 3 ijms-23-04223-t003:** Sequences of primers from six candidate genes used in qRT-PCR.

Gene ID (Encoded Protein)	Primer Sequence (5′ to 3′)
*Zm00001d051837* (Sucrose synthase 6)	F: ATGGACCACAGGTATCATTTCTCA R: TGGCATTGTGAACGCATAGTG
*Zm00001d051520* (MYB-related protein)	F: GCTGGAGAATTTGGAGAAGGAG R: CTGGAAGTAGGTGGAGACTGGG
*Zm00001d051829* (U-box domain-containing protein 44)	F: TCCTCAAGTGGTTCAGGGAGT R: CAGGTCAGCTATGGTGGGTATG
*Zm00001d051928* (Glutathione S-transferase family protein)	F: TTCACGACGCTCATCCGA R: CCTCAAGAGCGACTCCCTATC
*Zm00001d051687* (Rho GDP-dissociation inhibitor 1)	F: GAGATGCTCGGCACGTTCA R: AGGCGTAGTTGATCTCCAGGTA
*Zm00001d051847* (Nana2-like1)	F: GATGATGACGGGCGTGTATG R: CTTGAACAGGCGATGCGG
*ZmActin1* (Actin 1)	F: CGATTGAGCATGGCATTGTCA R: CCCACTAGCGTACAACGAA

## Data Availability

The data that support the findings of this study are available at NCBI under SRA PRJNA. Accession numbers can be found at Appendix A.

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
