# Peer review of "The Combination of Conventional QTL Analysis, Bulked-Segregant Analysis, and RNA-Sequencing Provide New Genetic Insights into Maize Mesocotyl Elongation under Multiple Deep-Seeding Environments"

_ijms, 2022, doi:10.3390/ijms23084223_

Round 1

Reviewer 1 Report

In addition to the comments on the marked up manuscript, the authors will need to have submitted the raw sequence reads to a publicly accessible data base such as NCBI's SRA database. 

Author Response

Thank you for your letter of – and for the referee’s comments concerning our manuscript, “The Combination of Conventional QTL Analysis, Bulked-Segregant Analysis, and RNA-sequencing Provide New Genetic Insights into Maize Mesocotyl Elongation under Multiple Deep-seeding Environments (ijms-1642070)”. We have carefully studied these comments and have made corresponding corrections to the manuscript, which we describe in detail below. We would like to re-submit the manuscript and that for possible publication on the Special Issue: “Molecular Research in Maizeof International Journal of Molecular Sciences. Thank you very much for your time and consideration.

Reviewer 1:

Too much info here: How many QTL from these studies were specifically linked to MES? Just focus on that the sake of the reader. Here and below.

Thanks for your positive comments. To make readers better understand and pay attention to the number of detected QTLs for deep-seeding traits, especially mesocotyl length (MES) based on different populations from previous studies. As suggested, the corresponding contents have been modified that “For example, Han et al. [17] and Liu et al. [5] identified 23 QTLs for MES based on B73 × Mo17 IBM Syn4 recombinant inbred lines (RILs; 243 lines)/IBM Syn10 doubled haploid (DH; 280 lines) population at a sowing depth of 12.5 cm, and six candidate genes controlling the deep-seeding tolerance of maize were identified: GRMZM2G059167 (MYB38), GRMZM2G133836 (GA insensitive dwarf 1, GID1), GRMZM2G139680 (peroxiredoxin family), GRMZM2G140633 (cell cycle regulator, cyclin delta-2), GRMZM2G151230 (cell number regulator 2, CNR2), and GRMZM2G098460 (drought-induced protein 1) in multiple QTLs regions [5].” in Lines 63-70 of the manuscript. We then have re-submitted the manuscript.

Thank you for your consideration.

Again, just give the numbers and candidate genes for MES.

Thanks for your positive comments. As suggested, the corresponding contents have been modified that “In another study, Chen et al. [19] identified three linked single nucleotide polymorphisms (SNPs) that were associated with MES in 165 inbred maize lines via genome-wide association studies (GWAS), and three candidate genes: GRMZM2G149580 (acyl carrier protein, ACP4), GRMZM2G037378 (plus-3 domian-contaning protein), and GRMZM2G078906 (FAD binding domain-containing protein), were further validated in these SNPs regions.” in Lines 70-75 of the manuscript. We then have re-submitted the manuscript.

Thank you for your consideration.

This is a difficult sentence to understand. Perhaps break it into two or three sentences?

Thanks for your positive comments. As suggested, the corresponding contents have been modified that “Moreover, compared to the normal sowing depth of 3 cm, the MES was increased by 169.11, 41.38, 233.77, and 192.78% in W64A, K12, F1 hybrid, and F2:3 families under 15 cm deep-seeding condition, meanwhile it was increased by 232.46, 124.14, 293.77, and 250.00% in these materials under 20 cm deep-seeding condition, respectively (Figure 1E).” in Lines 111-115 of the manuscript. We then have re-submitted the manuscript.

Thank you for your consideration.

Also, the fact that the F1 is actually bigger than the parent suggests that the trait is controlled by a limited number of dominant genes (and that there is likely some hybrid vigor involved).

Thanks for your positive comments. Yes, there was a complex heterosis of mesocotyl length (MES) from our experiment, namely, “By considering the F1 heterosis index (HI, 161.44%), relative heterosis (RH, 36.15%), mid-parent heterosis (MH, 61.44%), over-parent heterosis (OH, 15.05%), and F2:3 advantage reduction rate (ARR, -10.43%) of MES, we were able to accurately select the basic background of maize with long mesocotyls and tolerance to deep-seeding environments for our study (Figure 1F).” in Lines 344-348 of the manuscript.

In addition, “Single- environment mapping with composite interval mapping (CIM) was used to investigate the genetic control underlying MES improvement. Seven significant QTLs (P < 0.05) for MES in the 346 F2:3 population were identified under 3, 15, and 20 cm sowing conditions, and they distributed on chromosomes (Chr.) 1, 3, 4, 6, and 7, respectively (Table 1; Figure 3A-3B). For these identified QTLs, approximately 68.75, 6.25, 6.25, and 18.75% showed additive (A), partial-dominance (PD), dominance (D), and over-dominance (OD) effects, respectively (Figure 3C). These data showed that MES was regulated by both additive and non-additive effects under different sowing depths.” in Lines 166-173 of the manuscript,

Further analysis revealed that “To further determine the genetic basis of deep-seeding tolerance in maize, especially MES, we validated seven QTLs (four, seven, and five QTLs at 3, 15, and 20 cm depths, respectively) responsible for MES under three sowing environments by conventional QTL analysis with CIM. Analysis showed that 68.75% and 31.25% of QTLs exhibited both additive and non-additive effects, respectively. These results are consistent with previous findings [31]. It is clear that additive and non-additive effects, as well as the general and specific ability for combination, should be applied to improve the MES of maize under normal and deep-seeding environments. Furthermore, MES can be modified in early generations by simple backcrossing or single cross recombinations.” in Lines 336-344 of the manuscript. Therefore, MES may be not entirely controlled by a limited dominant genes.

Thank you for your consideration.

Authors should indicate which colors correspond to which populations.

Thanks for your positive comments. The legend illustrations of all maize populations were given in the Figure 1C. Therefore, there is no need repeat the pillar color for each maize population in the title of Figure 1C. By reading the Figure 1C, readers can intuitively understand the information of the corresponding maize populations in Figure 1B (Lines 129-139) of the manuscript.

Thank you for your consideration.

I would suggest using “influenced” rat than “caused” as the “cause” is likely a small number of regulator genes.

Thanks for your positive comments. As suggested, we have corrected the corresponding contents, namely “These data suggested that the differences of MES between W64A and K12 under different sowing environments may be influenced by multiple DEGs in specific KEGG pathways.” in Lines 239-241 of the manuscript. We then have re-submitted the manuscript.

Thank you for your consideration.

Hmmm…so the top 20 are not ranked by either q value or enrichment factor but reverse alphabetical??? Why? Also, a q-value of 1 is not significant at all. I would suggest ranking by q value lowest to highest that way. Those of us that have trouble distinguishing colors can still see the relationships and easily see which ontologies are significantly…

Thanks for your positive comments. As suggested, we have re-made the Figure 5F-5I and have replaced the Fig. 5 (Lines 242-255 of the manuscript). We then have re-submitted the manuscript.

Thank you for your consideration.

This should be in the results section along with the associated Figure 6.

Thanks for your positive comments. As suggested, the corresponding contents have been placed in results section of the manuscript, namely, “2.2. Framework of phenotype and physiological metabolisms of MES. We then selected two parents, an F1 hybrid, and two bulked-pools under 3 and 20 cm conditions to measure the level of lignin and eight phytohormones (Figure 2A). Unlike the normal sowing depth of 3 cm, their average content of lignin and gibberellic acid 3 (GA3) decreased by 38.07 and 71.77% under 20 cm deep-seeding condition, respectively; however, their average content indole-3-acetic acid (IAA), cis-zeatin (Cis-ZT), trans-zeatin (Trans-ZT), ABA, 24-epibrassinolide (EBR), JA, and SA increased by 82.78, 156.40, 112.73, 107.55, 170.71, 205.35, and 72.49%, respectively (Figure 2A). In addition, correlation analysis showed that MES had significantly positive correlation in terms of IAA, ABA, EBR, Cis-ZT, Trans-ZT, and SA content, and displayed significantly negative correlation with lignin and GA3 content (Figure 2B-2C). At the same time, a total of 28 significantly positive/negative correlations were identified among other nine physiological metabolisms (Figure 2B-2C). These data indicated that there were close correlation among MES, phytohormones level, and lignin content. Therefore, maize mesocotyl elongation is largely due to changes in these physiological metabolisms at 20 cm sowing depth.” (Lines 140-154). In addition, the Figure 6 has be placed in manuscript as Figure 2 (Lines 155-164). We then have re-submitted the manuscript.

Thank you for your consideration.

Why not GA? It would seem that it should be one of the premier hormones controlling elongation.

Thanks for your positive comments. Yes, gibberellic acid (GA) plays an important role in maize mesocotyl elongation under different sowing depths (Zhao, X.Q.; Zhong, Y.; Zhou, W.Q. Molecular mechanism of mesocotyl elongation induced by brassinosteroid in maize under deep-seeding stress by RNA-sequencing, microstructure observation, and physiological metabolism. Genomics 2021, 113, 3565-3581; Zhao, G.; Fu, J.; Wang, G.; Ma, P.; Wu, L.; Wang, J. Gibberellin-induced mesocotyl elongation in deep-sowing tolerant maize inbred line 3681-4. Plant Breed. 2010, 29, 87-91).

As suggested, we have re-added the data of GA content in different maize populations mesocotyls at 3 and 20 cm sowing depths, namely, “2.2. Framework of phenotype and physiological metabolisms of MES. We then selected two parents, an F1 hybrid, and two bulked-pools under 3 and 20 cm conditions to measure the level of lignin and eight phytohormones (Figure 2A). Unlike the normal sowing depth of 3 cm, their average content of lignin and gibberellic acid 3 (GA3) decreased by 38.07 and 71.77% under 20 cm deep-seeding condition, respectively; however, their average content indole-3-acetic acid (IAA), cis-zeatin (Cis-ZT), trans-zeatin (Trans-ZT), ABA, 24-epibrassinolide (EBR), JA, and SA increased by 82.78, 156.40, 112.73, 107.55, 170.71, 205.35, and 72.49%, respectively (Figure 2A). In addition, correlation analysis showed that MES had significantly positive correlation in terms of IAA, ABA, EBR, Cis-ZT, Trans-ZT, and SA content, and displayed significantly negative correlation with lignin and GA3 content (Figure 2B-2C). At the same time, a total of 28 significantly positive/negative correlations were identified among other nine physiological metabolisms (Figure 2B-2C). These data indicated that there were close correlation among MES, phytohormones level, and lignin content. Therefore, maize mesocotyl elongation is largely due to changes in these physiological metabolisms at 20 cm sowing depth.” (Lines 140-154). In addition, we have re-made and replaced the Figure 2 (Lines 155-164). We then have re-submitted the manuscript.

Thank you for your consideration.

 This statement does not match with the observation that the F1 hybrid had as good or better MES than the parent. Nor does it match with the finding of two well defined QTL on chromosomes 1 and 4 identified by two independent analyses. This discrepancy should be discussed in some detail.

Thanks for your positive comments. Yes, this statement is wrong. Therefore, we have restated the corresponding contents that “For these identified QTLs, we further identified four major QTLs for MES (Table 1).” in Lines 348-349 of the manuscript. We then have re-submitted the manuscript.

Thank you for your consideration.

The F2s a might or might not be heterozygous for the major loci controlling MES. Why not just choose one (or more) individuals from each F2:3 family from the phenotyping experiment that were strong (or weak) elongators rather than going back to the essentially unphenotyped F2?

Thanks for your positive comments. Generally, bulked-segregation analysis (BSA) was performed across F2 population to rapidly identify quantitative trait loci (QTLs) for some important quantitative traits, such as plant height in a 940 rice F2 population (Zhang, B.; Qi, F.X.; Hu, G.; Yang, Y.K.; Zhang, L.; Meng, J.H.; Han, Z.M.; Zhou, X.C.; Liu, H.Y.; Ayaad, M.; Xing, Y.Z. BSA-seq-based identification of a major additive plant height QTL with an effect equivalent to that of semi-dwarf 1 in a large rice F2 population. The Crop Journal 2021, 9, 1428-1437), leaf trichome formation in a 294 Chinese cabbage F2 population (Zhang, R.J.; Ren, Y.M.; Wu, H.Y.; Yang, Y.; Yuan, M.G.; Liang, H.N.; Zhang, C.W. Mapping of genetic locus for leaf trichome formation in Chinese cabbage based on bulked segregant analysis), and stalk architecture in a 313 F2/F2:3 maize population (Wang, X.Q.; Shi, Z.; Zhang, R.Y.; Sun, X.; Wang, J.D.; Wang, S.; Zhang, Y.; Zhao, Y.X.; Su, A.G.; Li, C.H.; Wang, R.H.; Zhang, Y.X.; Wang, S.S.; Wang, Y.D.; Song, W.; Zhao, J.R. Stalk architecture, cell wall composition, and QTL underlying high stalk flexibility for improved lodging resistance in maize. BMC Plant Biology 2020, 20, 515).

In addition, the F2:3 population was obtained by self crossing F2 plants, and using F2:3 population, the QTLs or stable QTLs for some important traits could be identified under different environments/conditions. Moreover, the corresponding traits of F2:3 populations can completely reflect the characteristics of the F2 population.

In the context of these facts, and according to the research (Wang, X.Q.; Shi, Z.; Zhang, R.Y.; Sun, X.; Wang, J.D.; Wang, S.; Zhang, Y.; Zhao, Y.X.; Su, A.G.; Li, C.H.; Wang, R.H.; Zhang, Y.X.; Wang, S.S.; Wang, Y.D.; Song, W.; Zhao, J.R. Stalk architecture, cell wall composition, and QTL underlying high stalk flexibility for improved lodging resistance in maize. BMC Plant Biology 2020, 20, 515). In this study, “According to the MES phenotypes of all F2:3 families under the 20 cm condition, we selected 30 and 30 families with the largest and shortest MES, respectively. Next, we found their corresponding F2 plants to establish two groups of plants: a “30 mixture F2 plants with the largest MES” bulked-pool (LM-pool) and a “30 mixture F2 plants with the shortest MES” bulked pool (SM-pool)” (Figure 1A). We also used individual W64A and K12 mesocotyls to establish parental pools for BSA-sequencing analysis.” in Lines 454-459 of the manuscript.

Thank you for your consideration.

What were the seeds “soaked in and for how long and at what temperature?

Thanks for your positive comments. In this study, the maize seeds were soaked at room temperature (25℃) for 24 h in 100 mL of distilled water. Namely, “First, the seeds of each F2:3 family, F1 hybrid, and two parental lines were soaked at room temperature (25℃) for 24 h in 100 mL of distilled water.” in Lines 462-464 of the manuscript.

Thank you for your consideration.

Most read counting programs can adjust for reads that map to multiple locations. There is no need to only select unique-mapped reads for quantization of gene expression.

Thanks for your positive comments. I agree with your perspective. Thus we have deleted the corresponding contents in the manuscript. We then have re-submitted the manuscript.

Thank you for your consideration.

Is there evidence that Actin1 is not differentially expressed in the MES between Varieties? It would seem to me that this gene might ell be a target gene impacting growth. Please provide some evidence that Actin1 one is not differentially expressed itself under the condition in this experiment. This information should be available in the RNAseq data that was collected.

Thanks for your positive comments. The ZmActin1 (Zm00001d010159; GRMZM2G126010; Actin1) was widely as the internal reference gene to analyze the expression level of candidate genes involving in maize mesocotyl elongation at different sowing depth soils via qRT-PCR. The related reports were Liu et al. (Liu, H.J.; Zhang, L.; Wang, J.C.; Li, C.S.; Zeng, X.; Xie, S.P.; Zhang, Y.Z.; Liu, S.S.; Hu, S.L.; Wang, J.H.; et al. Quantitative trait locus analysis for deep-sowing germination ability in the maize IBM Syn10 population. Front. Plant Sci. 2017, 8, 813), Chen et al. (Chen, F.Q.; Ji, X.Z.; Bai, M.X.; Zhuang, Z.L.; Peng, Y.L. Network analysis of different exogenous hormones on the regulation of deep-sowing tolerance in maize seedlings. Front. Plant Sci. 2021, 12, 739101), Zhao et al. (Zhao, X.Q.; Zhong, Y.; Zhou, W.Q. Molecular mechanisms of mesocotyl elongation induced by brassinosteroid in maize under deep-seeding stress by RNA-sequencing, microstructure observation, and physiological metabolism. Genomics 2021, 113, 3565-3581), and Zhao and Zhong (Zhao, X.Q.; Zhong, Y. 24-epibrassinolide mediated interaction among antioxidant defense, lignin metabolism, and phytohormones signaling promoted better cooperative elongation of maize mesocotyl and coleoptile under deep-seeding stress. Russ. J. Plant Physiol. 2021, 68, 1194-1207). Therefore, these evidences suggest that the internal reference gene of ZmActin1 selected for this study is correct, and we have modified the corresponding contents that “The 2−∆∆Ct method was used to determine the relative gene expression levels; these were normalized to the ZmActin1 gene [2,4,5].” in Lines 590-591 of the manuscript. We then have re-submitted the manuscript.

Thank you for your consideration.

In addition to the comments on the marked up manuscript, the authors will need to have submitted the raw sequence reads to a publicly accessible data base such as NCBI's SRA database. 

Thanks for your positive comments. As suggested, we have re-added the raw sequence reads to a publicly accessible data base such as NCBI's SRA database, namely “Data Availability Statement: The data that support the findings of this study are available at NCBI under SRA PRJNA.” in Lines 595-596 of the manuscript. Now NCBI is reviewing the data. We then have re-submitted the manuscript.

Because the deadline for my revision has come up, thus I will provide the number of NCBI's SRA database later.

Reviewer 2 Report

In this manuscript, the authors identified QTLs and candidate genes associated with mesocotyl length by QTL and RNA-sequencing analyses. The results are useful to maize researchers. However, I have a few concerns listed as follows
1. Start the abstract with a clear statement on the scope, relevance, and intention of the study, before describing the main results.
2. The Authors could also try to come up with a (graphical) model about their methodlogy and findings.
3. The language in the text should be polished
4. It is not clear as to whether the samples for qPCR are from the same RNA material as the transcriptome, should be preferably from a different one.
5. The first or final paragraph of the discussion should clearly describe the main conclusions of the work, their importance, and potential for further studies.   
6. Increase the font size and resolution in all the figures

Author Response

Thank you for your letter of – and for the referee’s comments concerning our manuscript, “The Combination of Conventional QTL Analysis, Bulked-Segregant Analysis, and RNA-sequencing Provide New Genetic Insights into Maize Mesocotyl Elongation under Multiple Deep-seeding Environments (ijms-1642070)”. We have carefully studied these comments and have made corresponding corrections to the manuscript, which we describe in detail below. We would like to re-submit the manuscript and that for possible publication on the Special Issue: “Molecular Research in Maizeof International Journal of Molecular Sciences. Thank you very much for your time and consideration.

Reviewer 2:

In this manuscript, the authors identified QTLs and candidate genes associated with mesocotyl length by QTL and RNA-sequencing analyses. The results are useful to maize researchers. However, I have a few concerns listed as follows.

Thanks for your positive comments. As suggested, we have further revised and improved the manuscript. We then have re-submitted the manuscript.

Thank you for your consideration.

  1. Start the abstract with a clear statement on the scope, relevance, and intention of the study, before describing the main results.

Thanks for your positive comments. As suggested, we have revised and improved the abstract section of the manuscript. Namely, “Mesocotyl length (MES) is an important trait that affects the emergence of maize seedlings after deep-seeding and is closely associated with abiotic stress. The elucidation of constitutive-QTLs (cQTLs) and candidate genes for MES, and tightly molecular markers are thus of great importance in marker-assisted selection (MAS) breeding. Therefore, the objective of this study was to perform detailed genetic analysis of maize MES across 346 F2:3 families, 30/30 extreme bulks of an F2 population, and two parents by conventional QTL analysis, bulked-segregation analysis (BSA), and RNA-sequencing when maize was sown at the depths of 3, 15, and 20 cm, respectively. QTL analysis identified four major QTLs in Bin 1.09, Bin 3.04, Bin 4.06-4.07, and Bin 6.01 under two or more environments, which explained 2.89 - 13.97% of the phenotypic variance within a single environment. BSA results revealed the presence of seven significantly linked SNP/InDel-regions on chromomes 1 and 4, and six SNP/InDel-regions and the major QTL of qMES4-1 overlapped and formed a cQTL, cQMES4, within the 160.98 - 176.22 Mb region. In total, 18,001 differentially expressed genes (DEGs) were identified across two parents by RNA-sequencing, and 24 of these genes were conserved core DEGs. Finally, we validated 15 candidate genes in cQMES4 to involve in cell wall structure, lignin biosyntheis, phytohormones (auxin, abscisic acid, brassinosteroid) signal transduction, circadian clock, and plant organ formation and development. Our findings provide a basis for MAS breeding and enhance our understanding of the deep-seeding tolerance of maize.” (Lines 11-27). We then have re-submitted the manuscript.

Thank you for your consideration.

  1. The Authors could also try to come up with a (graphical) model about their methodlogy and findings.

Thanks for your positive comments. In this study, our goals are : “(â…°) to identify QTLs controlling MES based on conventional QTL analysis across a 346 F2:3 population under sowing environments of 3, 15, and 20 cm, (â…±) to analyze linked BSA-regions responsible for MES via BSA using genome re-sequencing across “30 mixed F2 plants with the largest MES” bulked-pool (LM-­­pool), “30 mixed F2 plants with the shorest MES” bulked-pool (SM-pool), and their two parental inbred lines; and (â…²) to explore constitutive-QTLs (cQTLs; i.e., overlapping QTL regions associated with multiple deep-seeding tolerant traits from both the present research and previous studies), identify and mine the underlying candidate genes in corresponding cQTLs by combining RNA-Seq and quantitative real-time PCR (qRT-PCR) analyses across the two parent mesocotyls at sowing depths of 3 and 20 cm. Our findings provide a basis for further QTL fine mapping, MAS breeding, and functional studies of deep-seeding tolerance in maize.” in Lines 93-104 of the manuscript.

Then we have described and discussed the corresponding results in depth. In these regards, we have revised and improved the abstract section of the manuscript. Namely, “Mesocotyl length (MES) is an important trait that affects the emergence of maize seedlings after deep-seeding and is closely associated with abiotic stress. The elucidation of constitutive-QTLs (cQTLs) and candidate genes for MES, and tightly molecular markers are thus of great importance in marker-assisted selection (MAS) breeding. Therefore, the objective of this study was to perform detailed genetic analysis of maize MES across 346 F2:3 families, 30/30 extreme bulks of an F2 population, and two parents by conventional QTL analysis, bulked-segregation analysis (BSA), and RNA-sequencing when maize was sown at the depths of 3, 15, and 20 cm, respectively. QTL analysis identified four major QTLs in Bin 1.09, Bin 3.04, Bin 4.06-4.07, and Bin 6.01 under two or more environments, which explained 2.89 - 13.97% of the phenotypic variance within a single environment. BSA results revealed the presence of seven significantly linked SNP/InDel-regions on chromomes 1 and 4, and six SNP/InDel-regions and the major QTL of qMES4-1 overlapped and formed a cQTL, cQMES4, within the 160.98 - 176.22 Mb region. In total, 18,001 differentially expressed genes (DEGs) were identified across two parents by RNA-sequencing, and 24 of these genes were conserved core DEGs. Finally, we validated 15 candidate genes in cQMES4 to involve in cell wall structure, lignin biosyntheis, phytohormones (auxin, abscisic acid, brassinosteroid) signal transduction, circadian clock, and plant organ formation and development. Our findings provide a basis for MAS breeding and enhance our understanding of the deep-seeding tolerance of maize.” (Lines 11-27).

Furthermore, in order to let readers have a more intuitive understanding of the idea of this research, we have re-made and provided the model of the corresponding populations construction and QTL/BSA analysis, i.e., Figure 1A (Lines 129-139). We then have re-submitted the manuscript.

Thank you for your consideration.

  1. The language in the text should be polished.

Thanks for your positive comments. As suggested, the English language of the manuscript has been well modified by Charlesworth Author Services (https://www.cwauthors.com.cn/), and we provide the proof of English language polish. We then have re-submitted the manuscript.

Thank you for your consideration.

  1. It is not clear as to whether the samples for qPCR are from the same RNA material as the transcriptome, should be preferably from a different one.

Thanks for your positive comments. Yes, in this study, the mesocotyls samples of total RNA for qRT-PCR and transcriptome analysis are different. As suggested, we have revised and improved the corresponding contents that “We extracted total RNA from the mesocotyls of W64A, K12, F1 hybrid, LM-pool, and SM-pool under both 3 and 20 cm sowing conditions again, which were used for qRT-PCR analysis. The qRT-PCR was performed using a super real premix plus (SYBR Green) (Tiangen, Shanghai, China) in a total volume of 20 μL. The primers used to amplify the six candidate genes are listed in Table 3. The 2−∆∆Ct method was used to determine the relative gene expression levels; these were normalized to the ZmActin1 gene [2,4,5].” in Lines 586-591 of the manuscript. We then have re-submitted the manuscript.

Thank you for your consideration.

  1. The first or final paragraph of the discussion should clearly describe the main conclusions of the work, their importance, and potential for further studies.

Thanks for your positive comments. As suggested,  in the first paragraph of the discussion section of the manuscript, we have re-added the importance and potential for further studies of the work, i.e., “In the context of these facts, longer mesocotyl can as a desirable trait in maize, which is an asset for seedling emergence and establishment in adverse environments of top layer of the soil, particularly drought prone regions with less moisture, and the genetic insights into MES will provide a foundation for the genetic improvement and optimization of maize genotypes across different environments.” in Lines 330-334 of the manuscript. We then have re-submitted the manuscript.

In addition, in the final paragraph of the discussion section of the manuscript, we have re-added the importance and potential for further studies of the work, i.e., “In conclusion, these 15 candidate genes in cQMES4 region can be used as an excellent source of candidate genes for investigating the genetic control of maize seed germination and seedling growth under deep-seeding condition. Little is known about the specific roles and functions of these candidate genes in maize, and the Arabidopsis model can be used as a highly suitable model for investigating MES and mesocotyl elongation. In the future study, our research will focus on the functional verification and signal transduction pathways of these candidate genes.” in Lines 4439-445 of the manuscript. We then have re-submitted the manuscript.

Thank you for your consideration.

  1. Increase the font size and resolution in all the figures

Thanks for your positive comments. As suggested, we have re-made and provided the all figures to increase their font size and resolution. We then have re-submitted the manuscript and all figures, i.e., Figure 1 (Lines 129-139), Figure 2 (Lines 155-164), Figure 3 (Lines 184-188), Figure 4 (Lines 217-221), Figure 5 (Lines 242-255), Figure 6 (Lines 268-278).

Thank you for your consideration.

Round 2

Reviewer 1 Report

The authors have addressed all of my concerns. I still question the assertion that the few individuals with phenotypes that were tested in the F2:3 families are representative of the phenotype of the F2. If the F2 were heterozygous for one or more major alleles controlling the trait, the tested individuals in the F2:3 families might be homozygous for elongation alleles, heterozygous, or homozygous for alleles that produce short MES. Thus, phenotyping of the F2:3 may not provide accurate information on the phenotype of the F2. 

Also, unless there is data to the contrary, the fact that the F1 is as large (or slightly larger) than the long MES parent is due to dominance of the long MES phenotype (with perhaps some heterosis). If it were heterosis alone, acting on multiple loci with small effects, then one would expect to see the F1 as having an intermediate phenotype between the parents that was slightly longer than the average between the parents. Again, this is why I think that it is more likely that there are a very limited number of loci controlling the bulk of this trait- despite the association analyses suggesting as many as seven. It should also be noted that the QTL analysis was based on the phenotypes of representative members of the F2:3 families, and thus may not actually be an accurate representation of the F2s- thus the QTL analysis may also suffer from that fact and might not be as accurate as it should be.

That all said, if the authors agreed to note these possibilities in the discussion, I would agree that the manuscript is publishable.

Author Response

Thank you for your letter of – and for the referee’s comments concerning our manuscript, “The Combination of Conventional QTL Analysis, Bulked-Segregant Analysis, and RNA-sequencing Provide New Genetic Insights into Maize Mesocotyl Elongation under Multiple Deep-seeding Environments (ijms-1642070)”. We have carefully studied these comments and have made corresponding corrections to the manuscript, which we describe in detail below. We would like to re-submit the manuscript and that for possible publication on the Special Issue: “Molecular Research in Maizeof International Journal of Molecular Sciences. Thank you very much for your time and consideration.

Editor:

Your manuscript has been reviewed by experts in the field. Please find your manuscript with the referee reports at this link: https://susy.mdpi.com/user/manuscripts/resubmit/39413fe33799ba8629 87af8719dae8d9

Thanks for the positive comments of you and all reviewers for our manuscript. As suggested, we have carefully revised and improved the manuscript using the version of our manuscript at the above link. We then have re-submitted the manuscript.

Thank you for your consideration.

(I) Please revise your manuscript according to the referees’ comments and upload the revised file within 5 days.

Thanks for the positive comments of you and all reviewers for our manuscript. As suggested, we have carefully revised and improved the manuscript using the version of our manuscript at the above link. We then have re-submitted the manuscript.

Thank you for your consideration.

(II) Please use the version of your manuscript found at the above link for your revisions.

Thanks for the positive comments of you and all reviewers for our manuscript. As suggested, we have carefully revised and improved the manuscript using the version of our manuscript at the above link. We then have re-submitted the manuscript.

Thank you for your consideration.

(III) Any revisions made to the manuscript should be marked up using the “Track Changes” function if you are using MS Word/LaTeX, such that changes can be easily viewed by the editors and reviewers.

Thanks for the positive comments of you and all reviewers for our manuscript. As suggested, we have carefully revised and improved the manuscript, and all contents were modified by “Track Changes” function of MS Word. We then have re-submitted the manuscript.

Thank you for your consideration.

(IV) Please provide a short cover letter detailing your changes for the editors’ and referees’ approval.

Thanks for the positive comments of you and all reviewers for our manuscript. As suggested, we have carefully revised and improved the manuscript. In addition, we have prepared a detailed response letter to all reviewers for each point, and then have re-submitted the manuscript.

Thank you for your consideration.

Reviewer 1:

The authors have addressed all of my concerns. I still question the assertion that the few individuals with phenotypes that were tested in the F2:3 families are representative of the phenotype of the F2. If the F2 were heterozygous for one or more major alleles controlling the trait, the tested individuals in the F2:3 families might be homozygous for elongation alleles, heterozygous, or homozygous for alleles that produce short MES. Thus, phenotyping of the F2:3 may not provide accurate information on the phenotype of the F2. Also, unless there is data to the contrary, the fact that the F1 is as large (or slightly larger) than the long MES parent is due to dominance of the long MES phenotype (with perhaps some heterosis). If it were heterosis alone, acting on multiple loci with small effects, then one would expect to see the F1 as having an intermediate phenotype between the parents that was slightly longer than the average between the parents. Again, this is why I think that it is more likely that there are a very limited number of loci controlling the bulk of this trait- despite the association analyses suggesting as many as seven. It should also be noted that the QTL analysis was based on the phenotypes of representative members of the F2:3 families, and thus may not actually be an accurate representation of the F2s- thus the QTL analysis may also suffer from that fact and might not be as accurate as it should be.

Thanks for your positive comments. Yes, I totally agree with you. As suggested, we have carefully revised and improved the manuscript, namely “This data indicated that the significant elongation of mesocotyls determined the tolerance of maize to deep-seeding, thus mesocotyls play a key role during the emergence of maize seeds in deep layers of soil; and the mesocotyls of F1 hybrid was longer than both parents suggested that it might be controlled by a limit number of dominant genes, along with the hybrid vigor involved.” (Lines 115-119 of the manuscript). “In addition, we have to point out that MES phenotyping of the F2:3 families may not provide accurate information on the MES phenotype of the corresponding F2 population, thus the QTLs analysis in this study may not be as accurate. To more objectively evaluate the MES phenotype and map reliable QTLs, it is necessary to construct RIL or DH population for further study in the future.” (Lines 366-370 of the manuscript). We then have re-submitted the manuscript.

Thank you for your consideration.

That all said, if the authors agreed to note these possibilities in the discussion, I would agree that the manuscript is publishable.

Thanks for your positive comments. As suggested, we have discussed all these possibilities that you pointed out in Lines 115-119 and Lines 366-370 of the manuscript. We then have re-submitted the manuscript.

Thank you for your consideration.

Sincerely,

Xiaoqiang Zhao professor

State Key Laboratory of Aridland Crop Science, Gansu Agricultural University

E-mail: zhaoxq3324@163.com
